# Constructing multiple active sites in iron oxide catalysts for improving carbonylation reactions

Shujuan Liu[1,3], Teng Li[1,3], Feng Shi [1], Haiying Ma[1,2], Bin Wang[1], Xingchao Dai[1] & Xinjiang Cui [1] ✉

Surface engineering is a promising strategy to improve the catalytic activities of heterogeneous catalysts. Nevertheless, few studies have been devoted to investigate the catalytic behavior differences of the multiple metal active sites triggered by the surface imperfections on catalysis. Herein, oxygen vacancies induced $Fe_2O_3$ catalyst are demonstrated with different Fe sites around one oxygen vacancy and exhibited significant catalytic performance for the carbonylation of various aryl halides and amines/alcohols with CO. The developed catalytic system displays excellent activity, selectivity, and reusability for the synthesis of carbonylated chemicals, including drugs and chiral molecules, via aminocarbonylation and alkoxycarbonylation. Combined characterizations disclose the formation of oxygen vacancies. Control experiments and density functional theory calculations demonstrate the selective combination of the three Fe sites is vital to improve the catalytic performance by catalyzing the elemental steps of PhI activation, CO insertion and C-N/C-O coupling respectively, endowing combinatorial sites catalyst for multistep reactions.

Due to the economic and environmental advantages of non-noble metals present, the development of heterogeneous catalysts based on earth abundant transition metals is of substantial importance in catalysis[1-3]. However, compared with noble metal catalysts which generally catalyze reactions efficiently under mild conditions[4,5], low activity and selectivity has been a drag on the exploration of non-noble metal catalysts (NMCs)[6-8]. To achieve outstanding catalytic performance, several strategies such as constructing multi-metallic nanoparticles[9,10], doping heteroatoms[11-13], creating metal-support interaction[14-16], have been widely applied. Recently, Beller described a protocol to synthesize catalyst by immobilizing metal complexes on solid supports and subsequently pyrolysis under inert atmosphere, enhancing the NMCs catalytic performance on different reactions[17-19]. In addition, single atom catalysts have been attracted considerable attention on the synthesis of NMCs with exclusive catalytic properties[20,21]. Despite numerous achievements on NMCs synthesis, there are still of great significance to create active and selective NMCs.

Surface engineering was considered as another effective strategy to regulate the surface charge distribution and optimize the active sites, and further extend the functionalities of NMCs[22-24]. Surface imperfections such as oxygen vacancies ($O_{vac}$) are ubiquitous in metal oxides which can modify the physical and chemical properties of the oxide materials significantly, involving the catalytic performance dramatically in various reactions[25-27]. The $O_{vac}$ on oxide supports can not only tune the interaction of the metal and support but also serve as active sites directly with the metal active centers in the catalytic cycle, finely regulating the catalytic activity and selectivity[28]. The creation of $O_{vac}$ on $TiO_2$ benefits the formation of atomic interface between isolated Pt atom and surface $Ti^{3+}$ which facilitates the electron transfer between single Pt atoms and $Ti^{3+}$ sites, thereby enhancing the

[1]State Key Laboratory for Oxo Synthesis and Selective Oxidation, Lanzhou Institute of Chemical Physics, Chinese Academy of Sciences, No. 18, Tianshui Middle Road, Lanzhou 730000, China. [2]University of Chinese Academy of Sciences, No. 19A, Yuquan Road, Beijing 100049, China. [3]These authors contributed equally: Shujuan Liu, Teng Li. ✉e-mail: xinjiangcui@licp.cas.cn

photocatalytic hydrogen production[29]. Surface $O_{vac}$ on $Cu/CeO_2$ is beneficial to form the anti-sintering active sites by the synergistic effect with neighboring copper cluster, promoting the catalytic efficiency for the RWGS reaction and the catalyst stabilization even at high operating temperature[30]. By means of the $O_{vac}$ on $Ni@TiO_{2-x}$, electron density of surface Ni atoms increases by electronic migration from $TiO_{2-x}$ support to Ni atoms, forming the active architecture of $Ni^{\delta-}$-$O_V$-$Ti^{3+}$ and promoting the catalytic performance for WGS reaction[31]. In the presence of $O_{vac}$, grain refinement and spinel/perovskite heterostructure formation for perovskite oxides take place, leading to enhanced oxygen evolution reaction activity[32].

Although these achievements on the developments of $O_{vac}$ fabrications and applications in catalysis are reported, more attentions are paid on the electronic influence between $O_{vac}$ and adjacent single metal site. However, the formation of one oxygen vacancy normally causes the micro-environmental change of multiple metal sites. As we all know, the subtle variation in the active site architecture can affect catalytic performance significantly and even change the reaction pathway, thereby different elementary steps for a multistep reaction might be selectively determined by one of multiple metal sites. Thus, it is highly interesting to study the relationship between the catalytic reactivity with multiple metal sites.

Herein, active Fe sites with different catalytic performance are conducted by fabricating $O_{vac}$ on $Fe_2O_3$, which could serve as an ideal active catalyst for multistep carbonylation reaction of aryl halides and amines/alcohols with CO. The experiment results show that $Fe_2O_3$-$O_{vac}$ exhibits preeminent activity, selectivity, and durability in carbonylation of various aryl halides and nucleophiles, including amines, alcohols, drugs and chiral molecules derivatives. DFT calculations indicate that three different of Fe sites (denoted as **Fe1**, **Fe2** and **Fe3**) of $Fe_2O_3$-$O_{vac}$ are formed accompanying the formation of $O_{vac}$ (Fig. 1, Supplementary Fig. 1 and Fig. 2a, b). Importantly, the selective combination of these three Fe sites catalyzes different elementary reactions of PhI activation (**Fe1** and **Fe3** in Fig. 1b), CO insertion (**Fe1** and **Fe2** in Fig. 1c), C-N coupling (**Fe3**, **Fe1** and **Fe2** in Fig. 1d), respectively. The catalytic activity and selectivity of the carbonylation is significantly enhanced by the "combinatorial site catalysis" of these three Fe sites on $Fe_2O_3$-$O_{vac}$. This work reveals the catalytic differences of the multiple metal sites around $O_{vac}$ and improved catalytic performance is achieved by their combinatorial catalysis, providing a concept to the future rational catalyst design and activity enhancement of NMCs.

## Results

### Synthesis and characterization of catalysts

A series of $Fe_2O_3$ catalysts with different vacancies were synthesized via $NaBH_4$ reduction of the hydrothermally prepared $Fe_2O_3$[33–36], which were denoted as 0.5 $Fe_2O_3$-$O_{vac}$, 1.0 $Fe_2O_3$-$O_{vac}$ and 2.0 $Fe_2O_3$-$O_{vac}$ (0.5, 1.0 and 2.0 presented the molar ratio of $NaBH_4$: $Fe_2O_3$), as schematically illustrated in Fig. 2a. As shown in Supplementary Table 1, ICP-OES analysis for the starting material ($FeCl_3$) and $Fe_2O_3$-$O_{vac}$ catalyst revealed that the content of other metals such as Cu, Ni and Pd was below the limit of detection.

The X-ray diffraction (XRD) patterns (Supplementary Fig. 3) exhibited that $Fe_2O_3$ samples were formed with a high crystalline structure where (104) plane was dominant, which could be well indexed to the standard XRD pattern of hematite $Fe_2O_3$ (PDF #: 890599)[37]. $^{57}Fe$ Mössbauer spectroscopy is a very useful technique to explore the local magnetic behavior as well as the oxidation state of Fe atoms, the transmission Mössbauer spectra at room temperature for $Fe_2O_3$, 0.5 $Fe_2O_3$-$O_{vac}$ and 1.0 $Fe_2O_3$-$O_{vac}$ (Supplementary Fig. 4) only exhibited the single of the hematite $Fe_2O_3$, excluding the formation of the iron nanoparticles (Supplementary Table 3)[38], which was consistent with the XRD analysis. As shown in Fig. 2b–d, TEM images revealed the cube structure of $Fe_2O_3$ samples with 400 nm diameter. High-resolution TEM (HR-TEM) images in Fig. 2e–g provided information for the structure of $Fe_2O_3$ samples, where the lattice fringe spacing of 0.280 nm was observed mainly, corresponding to (104) lattice of the $Fe_2O_3$[37]. It was notable that the (104) lattice was detected for $Fe_2O_3$, 0.5 $Fe_2O_3$-$O_{vac}$ and 1.0 $Fe_2O_3$-$O_{vac}$, and no change on morphologies was observed after reduction by $NaBH_4$. However, increasing the molar ratio of $NaBH_4$: $Fe_2O_3$ to 2, the resulting 2.0 $Fe_2O_3$-$O_{vac}$ catalyst exhibited an obvious disruption of morphology and surface lattice, probably due to the over-reduction (Supplementary Fig. 5). X-ray photoelectron spectroscopy (XPS) was organized to study the composition of surface oxygen species and the charge distribution. All XPS spectra were charge corrected and referenced with adventitious carbon (284.8 eV). The survey spectra of three $Fe_2O_3$ samples were shown in Supplementary Fig. 6, which clearly revealed the coexistence of Fe, O, and C elements of three samples. The density of oxygen vacancy on these catalysts can also be deduced from O 1s spectra. As shown in Fig. 3a, three peaks can be deconvoluted from the O 1s profiles. The peaks at 529.8, 531.4, and 533.0 eV can be ascribed to surface lattice oxygen ($O_L$), surface $O_{vac}$ and other weakly bound

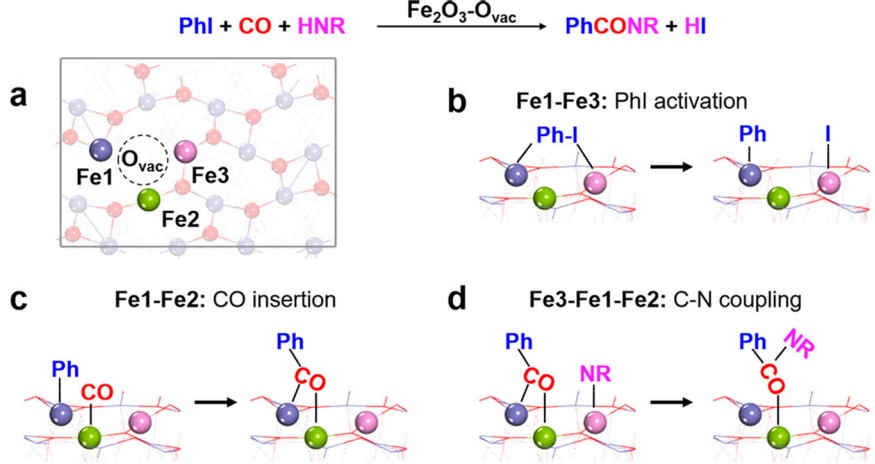

**Fig. 1 | Combinatorial sites catalysis of three different Fe sites (denoted as Fe1, Fe2 and Fe3) of $Fe_2O_3$-$O_{vac}$ caused by one $O_{vac}$. a** Architecture of three different Fe sites on $Fe_2O_3$-$O_{vac}$; (**b**) **Fe1-Fe3**: PhI activation; (**c**) **Fe1-Fe2**: CO insertion; (**d**) **Fe3-Fe1-Fe2**: C-N coupling. Note: $Fe_2O_3$-$O_{vac}$ represents $Fe_2O_3$(104)-$O_{vac}$ surfaces.

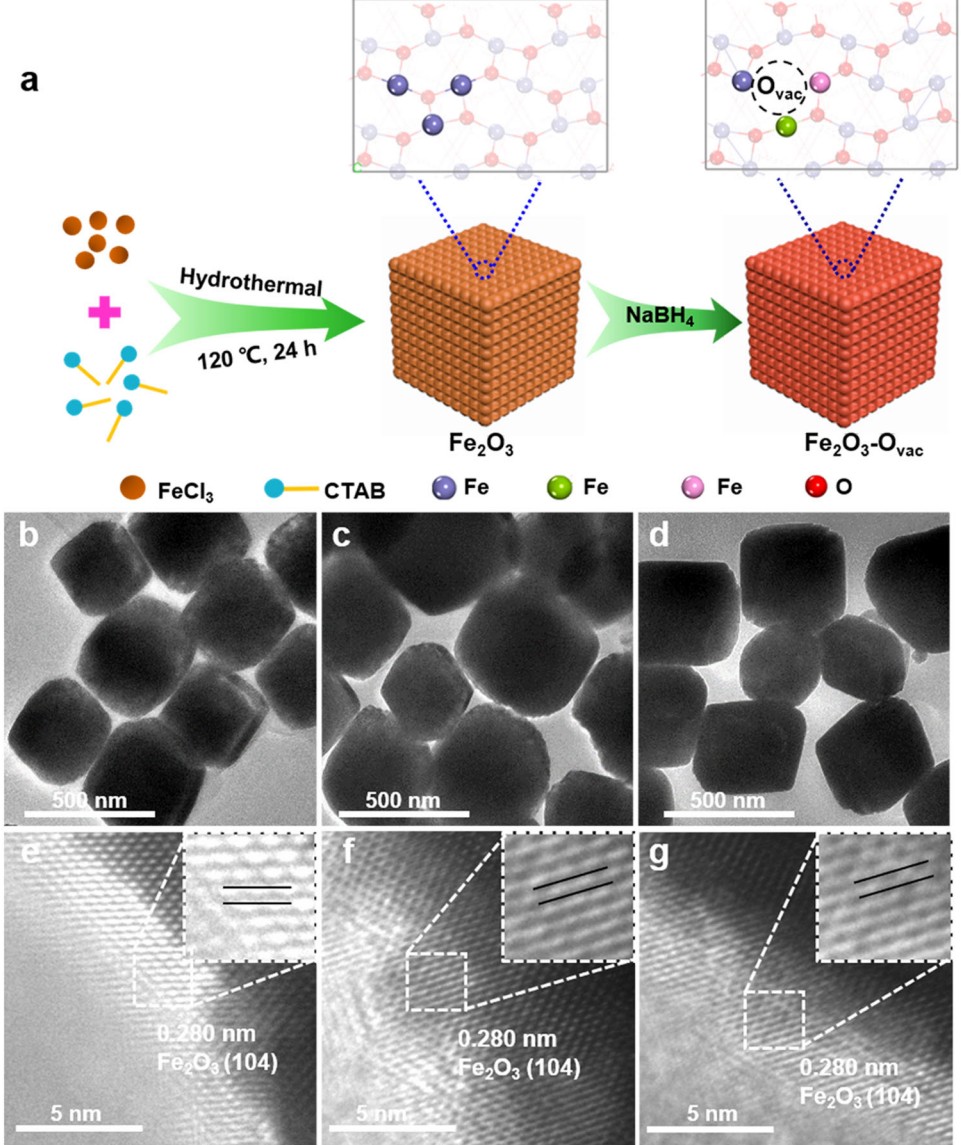

**Fig. 2 | Synthesis of the Fe$_2$O$_3$-O$_{vac}$ catalysts. a** Synthetic schematic for Fe$_2$O$_3$-O$_{vac}$; TEM images and high-resolution TEM (HR-TEM) images for Fe$_2$O$_3$ (**b**, **e**), 0.5 Fe$_2$O$_3$-O$_{vac}$ (**c**, **f**) and 1.0 Fe$_2$O$_3$-O$_{vac}$ (**d**, **g**).

oxygen species such as adsorbed molecular water and hydroxyl groups (O$_{OH}$)[39]. The density of O$_{vac}$ could be supposed as O$_{vac}$/ (O$_{vac}$ + O$_L$ + O$_{OH}$). As shown in Fig. 3a and Supplementary Table 3, the density of O$_{vac}$ was in the following order: 1.0 Fe$_2$O$_3$-O$_{vac}$ (26%) > 0.5 Fe$_2$O$_3$-O$_{vac}$ (18%) > Fe$_2$O$_3$ (13%), suggesting that the content of oxygen vacancies was gradually increased using larger NaBH$_4$ amounts. Note that the 2.0 Fe$_2$O$_3$-O$_{vac}$ showed lower content of O$_{vac}$, probably due to the damage of the morphology and surface lattice caused by the over-reduction (Supplementary Fig. 7). The content of surface hydroxyl groups was also analyzed in Supplementary Table 4, which showed that the content of surface hydroxyl groups of Fe$_2$O$_3$, 0.5 Fe$_2$O$_3$-O$_{vac}$ and 1.0 Fe$_2$O$_3$-O$_{vac}$ were almost similar, which revealed the catalytic efficiency was not affected by the surface hydroxyl groups. Moreover, two characteristic binding energy peaks accompanied by broad satellite peaks of the Fe 2p spectrum were observed at 710.9 eV (Fe 2p3/2) and 724.6 eV (Fe 2p1/2), respectively[40] (Supplementary Fig. 8). Additionally, the Fe 2p3/2 peaks in Fe$_2$O$_3$-O$_{vac}$ were slightly shifted to lower binding energies after the introduction of O$_{vac}$, which suggested that the valence state of Fe was partly decreased. These results

indicated that formation of O$_{vac}$ increased the electronic density of surface Fe in Fe$_2$O$_3$-O$_{vac}$.

The existence of O$_{vac}$ in crystals could be qualitatively determined by Electron Paramagnetic Resonance (EPR). Generally, a higher peak intensity in the EPR spectrum represented a higher concentration of O$_{vac}$[41] (Fig. 3b). All samples displayed EPR signals at the g-value of 2.003, indicating the trapping of electrons on O$_{vac}$. Interestingly, EPR signal of O$_{vac}$ for pristine Fe$_2$O$_3$ was observed, suggesting some O$_{vac}$ were fabricated during hydrothermal treatment process. The normalized EPR signal intensities were found to increase in the order of Fe$_2$O$_3$ < 0.5 Fe$_2$O$_3$-O$_{vac}$ < 1.0 Fe$_2$O$_3$-O$_{vac}$, significantly improved EPR intensity for 1.0 Fe$_2$O$_3$-O$_{vac}$ indicated the formation of abundant O$_{vac}$ via NaBH$_4$ reduction. To further investigate the change of O$_{vac}$ concentration in Fe$_2$O$_3$ samples, the samples were studied by thermogravimetry analysis (TGA)[42]. As shown in Fig. 3c, the total weight loss increased in the order of 1.0 Fe$_2$O$_3$-O$_{vac}$ (0.64%) < 0.5 Fe$_2$O$_3$-O$_{vac}$ (1.67%) < Fe$_2$O$_3$ (2.41%), confirming the presence of more O$_{vac}$ for 1.0 Fe$_2$O$_3$-O$_{vac}$ because of less weight loss. The weight losses observed in the H$_2$-TGA analysis

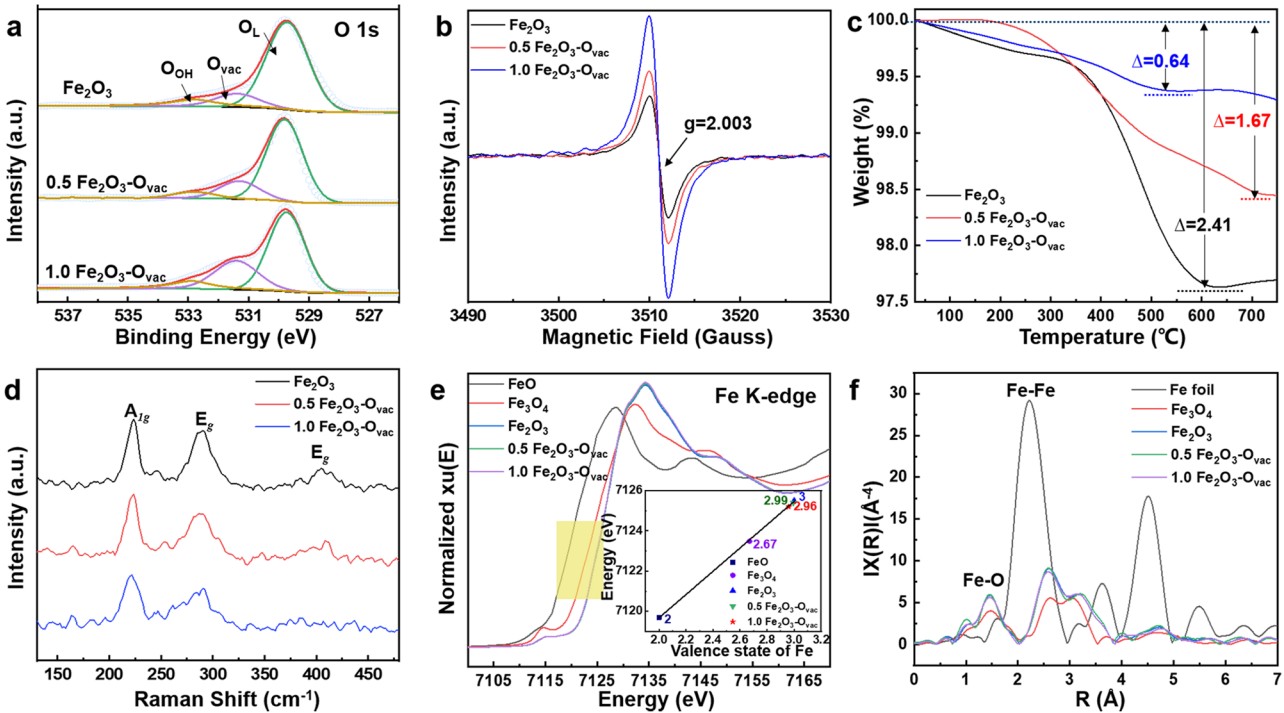

**Fig. 3 | Characterizations of catalysts. a** O 1 s XPS spectra; (**b**) EPR spectra; (**c**) Thermogravimetry analysis (TGA); (**d**) Raman spectra; (**e**) XANES spectra of Fe K-edge; (**f**) EXAFS Fourier Transformed (FT) spectra.

resulted from the desorption of water formed by the reduction reaction ($Fe_2O_3 + H_2 \rightarrow Fe^{2+}/Fe^0 + H_2O$)[43] in Supplementary Fig. 9, and two separated reduction stages can be distinguished. The first region between 250 and 500 °C corresponded to the reduction of Fe(III) to Fe(II) where the oxygen vacancies were mainly produced. The weight loss decreased in sequence of 1.0 $Fe_2O_3$-$O_{vac}$, 0.5 $Fe_2O_3$-$O_{vac}$ and $Fe_2O_3$, indicating the more oxygen vacancies were formed in 1.0 $Fe_2O_3$-$O_{vac}$ sample.

Raman spectra also revealed the typical peaks of hematite $Fe_2O_3$ for all samples (Fig. 3d)[30]. The peaks located at 221.7 $cm^{-1}$ was assigned to the Fe-O symmetric stretching vibrations ($A_{1g}$ mode), and the two peaks at about 287.9, and 408.8 $cm^{-1}$ were attributed to the Fe-O symmetric bending vibrations ($E_g$ mode). Compared with $Fe_2O_3$ and 0.5 $Fe_2O_3$-$O_{vac}$, 1.0 $Fe_2O_3$-$O_{vac}$ showed red-shifted and broadened peaks, which demonstrated the formation of $O_{vac}$ and large amounts of $O_{vac}$ caused more disrupted lattice of Fe-O bonding. The $O_2$-TPD was further conducted to study the change of $O_{vac}$ because $O_2$ released in low-temperature regions (<400 °C) was labile oxygen species[44]. Oxygen species desorbed from the catalyst surface below 400 °C increased in the order of $Fe_2O_3$ < 0.5 $Fe_2O_3$-$O_{vac}$ < 1.0 $Fe_2O_3$-$O_{vac}$ (Supplementary Fig. 10a), indicating that 1.0 $Fe_2O_3$-$O_{vac}$ possessed highest $O_{vac}$ concentration, in consist with the XPS, EPR, TGA, and Raman results. $H_2$-TPR experiments showed 1.0 $Fe_2O_3$-$O_{vac}$ consumed less $H_2$ than $Fe_2O_3$, 0.5 $Fe_2O_3$-$O_{vac}$ (Supplementary Fig. 10b and Supplementary Table 5), indicating more reliable surface oxygen atoms were removed during the $NaBH_4$ reduction, thereby generating more $O_{vac}$.

XAFS spectroscopy was utilized to probe detailed structure information such as the coordination environment[45]. Figure 3e showed the Fe K-edge X-ray absorption near-edge structure (XANES) spectra of the $Fe_2O_3$-$O_{vac}$ samples compared with FeO, $Fe_3O_4$ and $Fe_2O_3$ as references. The absorption threshold for the $Fe_2O_3$-$O_{vac}$ was close to that of $Fe_2O_3$, and the oxidation states of 0.5 $Fe_2O_3$-$O_{vac}$ and 1.0 $Fe_2O_3$-$O_{vac}$ were fitted and the average valence state calculated by area integration method[46] was approximately 2.99+ and 2.96+, the related stoichiometry of 0.5 $Fe_2O_3$-$O_{vac}$ and 1.0 $Fe_2O_3$-$O_{vac}$ were $Fe_2O_{2.99}$ and

$Fe_2O_{2.96}$, which was agree with the results of ICP-OES. Compared with reference samples, a major peak of Fe K-edge centered at around 1.5 Å was found in the 1.0 $Fe_2O_3$-$O_{vac}$ (Fig. 3f), which can be assigned to Fe-O coordination and no characteristic peaks of Fe-Fe coordination contribution were detected, indicating the absence of metallic Fe species. Further Extended X-ray absorption fine structure (EXAFS) fitting results revealed that the Fe-O coordination number of $Fe_2O_3$, 0.5 $Fe_2O_3$-$O_{vac}$ and 1.0 $Fe_2O_3$-$O_{vac}$ were 6.0, 5.8 and 5.4, respectively, which decreased with the increased dosage of $NaBH_4$ (Supplementary Table 5). The changes of Fe-O coordination number indicated the creation of $O_{vac}$ of $Fe_2O_3$ by the reduction treatment and more $O_{vac}$ were formed for 1.0 $Fe_2O_3$-$O_{vac}$ catalyst.

## Catalytic performance of the aminocarbonylation of iodobenzene

Carbonylation of aryl halides over transition metal-based catalysts is well known as a direct route to synthesis carbonyl compounds, such as carboxylic acids, amide, ester, and ketones[47–50]. Although many research works focused on the development of various Pd-based catalysts for amides and esters production (Supplementary Table 7), non-noble metal heterogeneous systems which efficiently catalyze the aminocarbonylation and alkoxycarbonylation are rare reported so far.

The catalytic performance of the prepared catalysts was examined using the carbonylation of iodobenzene and morpholine as benchmark reaction (Table 1). The triethylamine was generally added to neutralize the hydrogen halide formed during the reaction[51]. Using pristine $Fe_2O_3$, low catalytic activity with 15% yield was attained (entry 1). Surprisingly, the $NaBH_4$ reduction treatment led to a great increase in the catalytic performance with 50% yield obtained by 0.5 $Fe_2O_3$-$O_{vac}$ (entry 2). In the presence of 1.0 $Fe_2O_3$-$O_{vac}$, extremely high activity was obtained with 97% yield (entry 3). However, 2.0 $Fe_2O_3$-$O_{vac}$ led to quite low activity with only 22% yield (entry 4). The poor performance was attributed to the disruption and collapse of the surface-active sites caused by the over-reduction. These results indicated that the amount of $O_{vac}$ was vital to the catalytic activity. Moreover, control experiments where the iron nanoparticles and physical mixture of iron nanoparticles and 1.0

Fe₂O₃-O_{vac} catalysts were conducted, much lower yields of amide were obtained by adding iron nanoparticles (entries 2–4, Supplementary Table 2). To exclude the effect of Cu, Ni and Pd metals in the catalytic performance, Fe₂O₃-O_{vac} catalysts containing 250 ppm of Cu, Ni and Pd

### Table 1 | Catalyst screening and reaction conditions optimization[a]

| Entry | Catalyst | T/°C | CO/MPa | Solvent | Yield.[b]/% |
|---|---|---|---|---|---|
| 1 | Fe₂O₃ | 160 | 1 | 1,4-Dioxane | 15 |
| 2 | 0.5 Fe₂O₃-O_{vac} | 160 | 1 | 1,4-Dioxane | 50 |
| 3 | 1.0 Fe₂O₃-O_{vac} | 160 | 1 | 1,4-Dioxane | 97 |
| 4 | 2.0 Fe₂O₃-O_{vac} | 160 | 1 | 1,4-Dioxane | 22 |
| 5 | 1.0 CuO-O_{vac} | 160 | 1 | 1,4-Dioxane | 10 |
| 6 | 1.0 V₂O₅-O_{vac} | 160 | 1 | 1,4-Dioxane | 17 |
| 7 | 1.0 ZrO₂-O_{vac} | 160 | 1 | 1,4-Dioxane | 26 |
| 8 | 1.0 Fe₂O₃-O_{vac} | 160 | 1 | Methanol | 18 |
| 9 | 1.0 Fe₂O₃-O_{vac} | 160 | 1 | CH₃CN | 93 |
| 10 | 1.0 Fe₂O₃-O_{vac} | 160 | 1 | THF | 81 |
| 11 | 1.0 Fe₂O₃-O_{vac} | 160 | 1 | Toluene | 93 |
| 12 | 1.0 Fe₂O₃-O_{vac} | 160 | 1 | n-Octane | 42 |
| 13 | 1.0 Fe₂O₃-O_{vac} | 160 | 0.1 | 1,4-Dioxane | 1 |
| 14 | 1.0 Fe₂O₃-O_{vac} | 160 | 0.5 | 1,4-Dioxane | 48 |
| 15 | 1.0 Fe₂O₃-O_{vac} | 160 | 2 | 1,4-Dioxane | 17 |
| 16 | 1.0 Fe₂O₃-O_{vac} | 160 | 3 | 1,4-Dioxane | 7 |
| 17 | 1.0 Fe₂O₃-O_{vac} | 145 | 1 | 1,4-Dioxane | 11 |
| 18[c] | 1.0 Fe₂O₃-O_{vac}−450 | 160 | 1 | 1,4-Dioxane | 13 |

[a]Reaction conditions: **1a** (1.5 mmol), **2a** (1.0 mmol), Et₃N (2.0 mmol), CO (1.0 MPa), catalyst (80 mg), solvent (2 mL), 160 °C (reaction temperature), 24 h.
[b]Yields were determined by GC-MS.
[c]450 °C calcined in air for 5 h.

were prepared and their catalytic activity were studied respectively. As shown in Supplementary Table 2, the addition of Cu and Ni metals in the Fe₂O₃-O_{vac} catalyst decreased the catalytic activity, while the Pd exhibited negligible effect (entries 5–7). For comparison, 1.0 CuO-O_{vac}, 1.0 V₂O₅-O_{vac} and 1.0 ZrO₂-O_{vac} were also examined, lower yields were obtained (entries 5–7). Various solvents including methanol (CH₃OH), acetonitrile (MeCN), tetrahydrofuran (THF), toluene and n-octane were tested (entries 8–12) but no improved yields obtained. The CO pressure was also investigated and the yields increased gradually with the increase of CO pressure below 1 MPa, but dramatically declined at CO pressure higher than 2 MPa, probably ascribing to the inhibition of activation of PhI by the favorable CO adsorption at high CO pressure (entries 13–16). In addition, low reaction temperature was unfavorable for the catalytic conversion (entry 17).

To further elucidate surface vacancies-mediated catalysis, 1.0 Fe₂O₃-O_{vac} was treated at 450 °C in air for 5 h. After calcination, the obtained 1.0 Fe₂O₃-O_{vac}−450 catalyst was immediately used for aminocarbonylation of iodobenzene under the identical conditions, and only 13% yield was obtained (Table 1, entry 18). O 1s spectra showed that the surface O_{vac} fractions were effectively reduced from 26% in 1.0 Fe₂O₃-O_{vac} to 14% in 1.0 Fe₂O₃-O_{vac}−450 (Supplementary Fig. 11). And the TGA curves of 1.0 Fe₂O₃-O_{vac}−450 coincided exactly with the pristine Fe₂O₃ (Supplementary Fig. 12). Based on these results, the decreased catalytic activity could only be attributed to the decreased concentration of surface O_{vac} after annealing at 450 °C in air.

The quantitative connection between the content of O_{vac} and catalytic performance was investigated and the yield of **3a** was plotted against their content of O_{vac} for Fe₂O₃, 0.5 Fe₂O₃-O_{vac} and 1.0 Fe₂O₃-O_{vac} (Fig. 4a). A good linear correlation was observed, clarifying the significance of the surface O_{vac} on their catalytic performance. The recycling experiment of the 1.0 Fe₂O₃-O_{vac} catalyst showed that 93% yield of **3a** still remained after five batches of recycling (Fig. 4b), demonstrating the excellent reusability of 1.0 Fe₂O₃-O_{vac}. The stable nature of used 1.0 Fe₂O₃-O_{vac} catalyst was further characterized. Compared to the fresh sample, the morphology of used Fe₂O₃-O_{vac} was

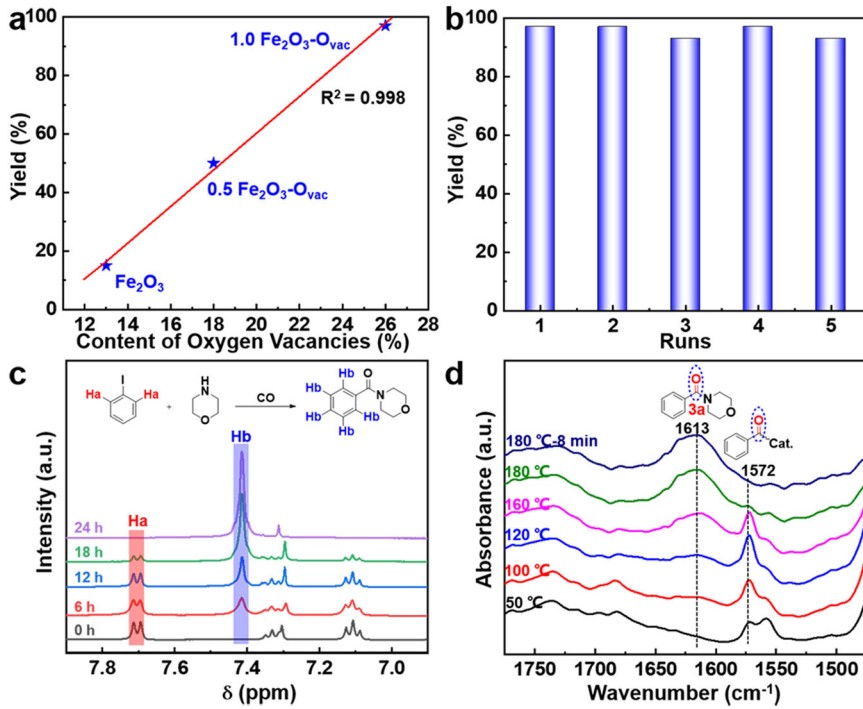

**Fig. 4 | Control experiment. a** The linear relationship between the content of O_{vac} and yield; (**b**) Recycling experiment; (**c**) Quasi in situ ¹H NMR spectra of the aminocarbonylation of iodobenzene; (**d**) In situ DRIFT spectra for the aminocarbonylation of iodobenzene over 1.0 Fe₂O₃-O_{vac}.

basically unchanged (Supplementary Fig. 13) and Raman characteristic peaks were observed no shifts of the Fe-O stretching vibration band (Supplementary Fig. 14). Besides, the content of $O_{vac}$ in the 1.0 $Fe_2O_3$-$O_{vac}$ after recycling was close to the fresh one (Supplementary Fig. 15). These results revealed the solid structure stability of 1.0 $Fe_2O_3$-$O_{vac}$ during the reactions. The quasi in situ $^1$H NMR spectra were used to monitor the aminocarbonylation reaction process. As shown in Fig. 4c, the signals corresponding to the -$C_6H_5$- group (Ha in Fig. 4c) in iodobenzene decreased over time, while the peaks corresponding to the -$C_6H_5$- group (Hb in Fig. 4c) of **3a** increased simultaneously.

In situ DRIFTS analysis was performed to uncover the dynamic catalytic behavior of the 1.0 $Fe_2O_3$-$O_{vac}$ and the possible reaction intermediates in the aminocarbonylation of iodobenzene process (Fig. 4d). The characteristic peak of the PhCO* group, which was formed during the CO insertion elementary reaction, appeared immediately at 1572 $cm^{-1}$ [52] at 50 °C, and then reached maximum intensities at about 120 °C and gradually declined until disappeared. Simultaneously, the signal intensity of stretching vibration of -C = O- of the target amide located at 1613 $cm^{-1}$ was intensified, which exhibited the generation of **3a** as the reaction proceeding. These studies confirmed the formation of PhCO* intermediate and in situ transformation into the desired products during the reaction using 1.0 $Fe_2O_3$-$O_{vac}$. Control experiments were conducted to study the possibility of radical mechanism (Supplementary Table 8). Under the standard conditions, 99% amide (**3a**) were obtained by adding 2,2,6,6-tetra-methylpiperidine-1-oxyl (TEMPO, a radical scavenger) and equivalent molar amount of morpholine, excluding the radical mechanism, and we proposed the reaction process was divided into three elementary reaction steps: the PhI activation (step **I**), the CO insertion (step **II**) and the C-N coupling (step **III**) (Fig. 5a)[49].

## DFT calculations

Density functional simulation was performed to elucidate the activity of 1.0 $Fe_2O_3$-$O_{vac}$. Assisted by the formation of $O_{vac}$, electrons were redistributed to the surrounded Fe atoms where the bader charge was reduced from 1.40 to 1.26 for **Fe1** and **Fe3**, and from 1.69 to 1.29 for **Fe2** (Supplementary Fig. 1). In addition, the coordination environment was also changed after removal of surface oxygen as shown in Supplementary Fig. 1. The influence of the variation of electronic and coordinated circumstances on the catalytic mechanism was studied in detail by DFT calculations, the whole potential energy surface (PES) for **3a** synthesis was depicted in Fig. 5b.

In the presence of $Fe_2O_3$-$O_{vac}$, iodobenzene was adsorbed at surface with adsorption energy of −1.02 eV (**IM1**) (Supplementary Fig. 2c). The PhI activation (step **I**) was occurred on $Fe_2O_3$-$O_{vac}$ with the energy barrier and reaction energy of 0.54 eV (**TS1**) and −0.55 eV (Supplementary Fig. 16). The hydroxyl groups, which might exist on the surface, had little effect on the energy barrier (0.55 eV on the surface with hydroxyl groups) but increased the reaction energy (−0.06 eV) of PhI dissociation, indicating that surface hydroxyl groups were not beneficial for C-I bond decomposition (**IM1'** → **TS1'** → **IM2'**, Supplementary Fig. 16). Therefore, PhI activation preferred to occurred on the vacant site without surface hydroxyl groups. After C-I cleavage, the phenyl group of iodobenzene was spontaneously moved to the **Fe1** site by the formation of **Fe1**-C intermediate (**IM2**). Afterwards, the CO insertion (step **II**) was triggered by CO adsorption on **Fe2** site with the adsorption energy of −0.66 eV (**IM3**) (Supplementary Fig. 17). Subsequently, CO was inserted into the **Fe1**-C bond with the generation of the acyl intermediate (PhCO*) (**IM4**), which was confirmed by the DRIFTS measurement (Fig. 4d), with the energy barrier of 1.04 eV (**TS2**) and exergonic by 0.75 eV (Supplementary Fig. 18). Note that the formed PhCO* was adsorbed on **Fe1** and **Fe2** where C and O sites were bonded with **Fe1** and **Fe2** atom, respectively. This simulated structure (**IM4**) was beneficial for the amide formation (step **III**) which started with the adsorption of morphine [HNR] on **Fe3** site (**IM5**). Next, iodine was

displaced by NR* with the formation of HI, the formed NR* attacked C site of PhCO* with the barrier of 0.77 eV (**TS3**) and strong exothermic by 1.29 eV (Supplementary Fig. 18), leading to the formation of the desired amide product **3a** (**IM6**).

Clearly, C-I bond activation was the highest point among the PES with respect to the **IM1**, indicating that step **I** determined the apparent barrier of the reaction. Since the energy barrier for PhI activation on $Fe_2O_3$-$O_{vac}$ (0.54 eV) was lower than that on bare $Fe_2O_3$ (0.66 eV, Supplementary Fig. 16), the reaction was more likely to be triggered on $Fe_2O_3$-$O_{vac}$. Further studies revealed the change of apparent barrier for C-I scission was caused by the different adsorbed geometries where the iodobenzene was parallel adsorbed on $Fe_2O_3$-$O_{vac}$, completely different from the tilted adsorption on bare $Fe_2O_3$ (Supplementary Fig. 2c, d). Differential charge density plots for these two adsorbed states (**IM1** and **IM1″**, Fig. 5c, d) before C-I cleavage were explored. Obviously, electrons were accumulated at the zone between the phenyl group and flawed surface in **IM1** but no such phenomenon was observed in **IM1″**, indicating the strong interaction between phenyl and adsorbent. This configuration of **IM1** benefited the electrons transfer from **Fe1** to $C_6H_5$ group, stabilizing the phenyl motif during the PhI activation as shown in Fig. 5e. To clarify the relative correspondence of the C-I bond activation with the surface electron transfer ability, the d-band center was depicted (Fig. 5f, g). Notably, the **Fe1** and **Fe3** of $Fe_2O_3$-$O_{vac}$ displayed a positive value (−1.42 eV) than that of bare $Fe_2O_3$ (−2.81 eV). The closer of d-band center to fermi level indicated the preferable charge donation from the active site to absorbent, elucidating the easier C-I bond activation by $Fe_2O_3$-$O_{vac}$[53]. Homogeneous complexes where transition metals were generally in lower oxidation states initiated the reaction by the rate-determined activation of aryl halides[54,55], However, the CO migratory insertion (**TS2**) was the rate-determining step in our 1.0 $Fe_2O_3$-$O_{vac}$ system which was different from transition metal process. DFT calculations and experiments results confirmed that the "combinatorial site catalysis" of three Fe sites induced $O_{vac}$ covered three different elementary reaction steps of the aminocarbonylation of iodobenzene, endowing significant improvement of the catalytic performance.

## Reaction system scoping

The scope and limitation of the aminocarbonylation of various aryl halides and amines into the corresponding amides were conducted over 1.0 $Fe_2O_3$-$O_{vac}$. As shown in Fig. 6, the aminocarbonylation of aryl iodide was investigated using various amines as the starting materials. Cyclic and aliphatic acyclic secondary amines with steric hindrance were well tolerated and 91–97% yields were obtained (**3a**-**3d**). In addition to secondary amines, both aliphatic and cyclic primary amines were well tolerated and converted to the desired amides in 82–99% yields (**3e**-**3m**). Interestingly, oleylamine was successfully transformed to the corresponding amide (**3n**) in 85% yield with the preservation of the unsaturated C = C bond. Moreover, amide **3o** was obtained in 83% yield using benzylamine as substrate. The carbonylation of aromatic amines with different functional groups occurred successfully to afford the desired amides in 91–99% yields (**3p**-**3r**). Furthermore, different aryl halides were tested on the aminocarbonylation with morpholine. Generally, both electron-donating and electron-withdrawing groups in aryl halides were well applicable under the identical conditions, affording the desired products in 71–99% yields. Compared with iodobenzene, 97% yield was achieved with *p*-Me substituted aryl iodides while the yield decreased to 84% when substituting by *p*-OMe (**3s** and **3t**). As expected, 1-naphthyl iodide also underwent this transformation smoothly, giving the desired product in 99% yield (**3u**). Importantly, bromobenzene with *p*-Cl, *p*-F and *p*-NO$_2$ substitutes were also selectively converted to the corresponding products in good to excellent yields (71–87%) after prolonging the reaction time (**3v**-**3aa**).

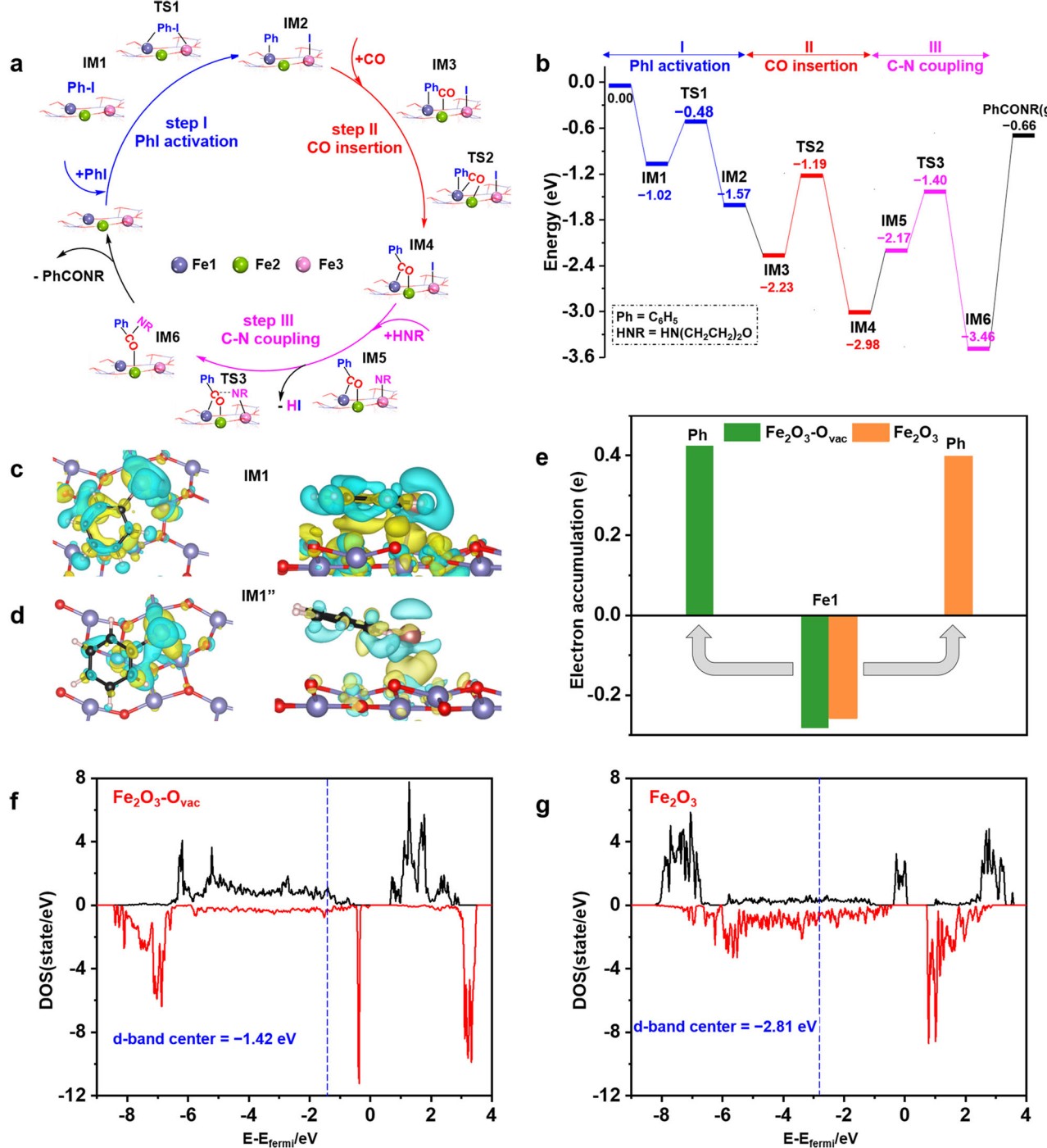

**Fig. 5 | DFT calculations. a** The reaction pathway for the aminocarbonylation of iodobenzene by 1.0 Fe₂O₃-O_vac; (**b**) Energy profiles of PhI aminocarbonylation over Fe₂O₃-O_vac surface; (**c, d**) Differential charge density plots (isosurface value of 0.008 eV/Å³; cyan, charge depletion; yellow, charge accumulation) of PhI on Fe₂O₃-O_vac and Fe₂O₃ surfaces (top view (left) and side view (right)); (**e**) Charge transfer upon the phenyl group at **Fe1**; (**f, g**) Density of states of **Fe1** and **Fe3** atoms from Fe₂O₃-O_vac and Fe₂O₃ surfaces (black line was spin-up, red line was spin-down; d-band center was calculated and the value presented was the average of spin-up and spin-down d-band centers). Note: Fe₂O₃ and Fe₂O₃-O_vac represent Fe₂O₃(104) and Fe₂O₃(104)-O_vac surfaces.

Encouraged by the outstanding catalytic results on the aminocarbonylation of aryl halides with amines, we subsequently studied the alkoxycarbonylation of aryl halides with alcohols using 1.0 Fe₂O₃-O_vac system. As shown in Fig. 7, a series of linear and branched primary alcohols were successfully converted to the corresponding benzoates with iodobenzene in excellent yields (**5a-5l**). When secondary alcohols, such as 2-butanol, 2-pentanol, cyclohexanol, cyclooctanol and cyclododecanol, were used as substrates, the transformations were efficiently conducted with

87–99% yields obtained (**5m-5q**). Note that the alkoxycarbonylation of aryl iodides with phenols was also catalyzed, providing phenyl benzoate in 99% yield (**5r**). Afterwards, the alkoxycarbonylation of various aryl halides was investigated using ethanol as starting material. Aryl halides bearing either electron-withdrawing or electron-donating groups were well tolerated, affording the corresponding benzoates in 73–96% yields (**5s-5z**). 1-Naphthyl iodide also offered 90% yield under the 1.0 Fe₂O₃-O_vac system (**5u**). Remarkably, 1-fluoro-4-bromobenzene was converted

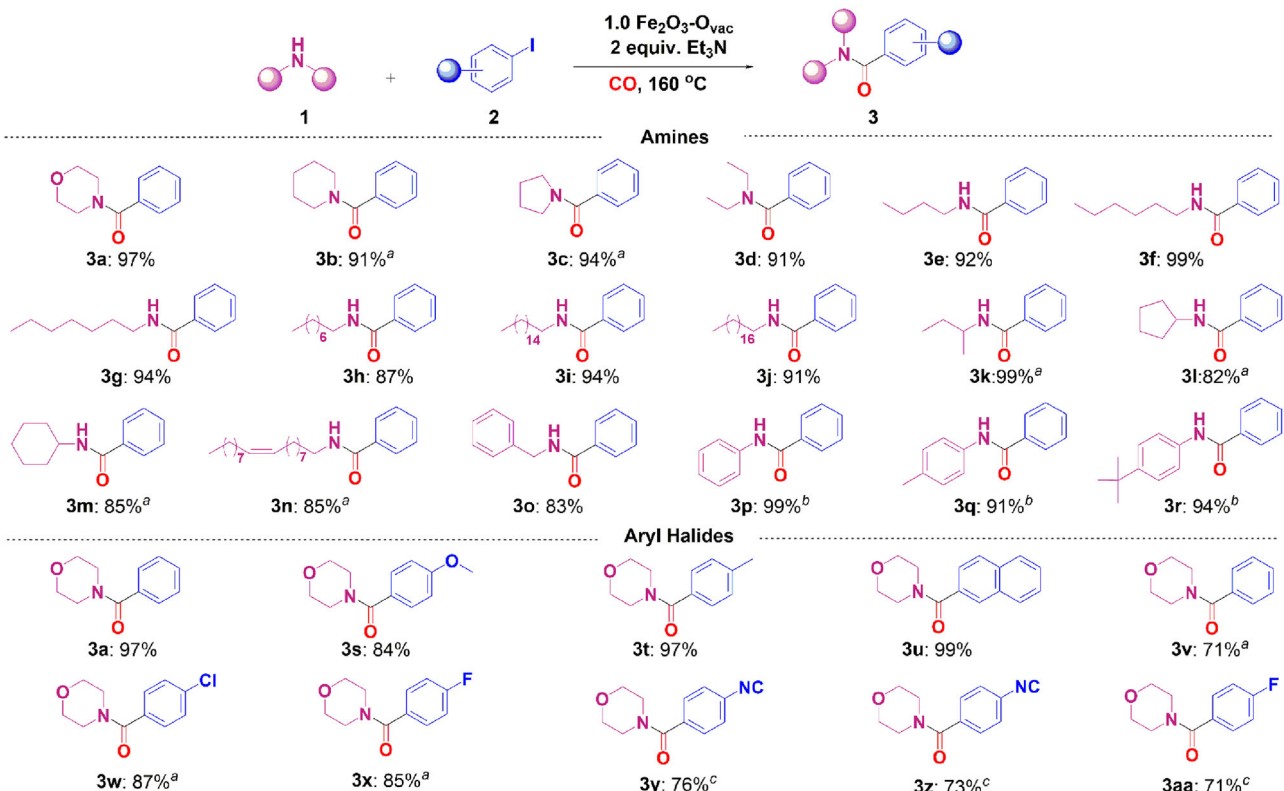

**Fig. 6 | Substrate scope of aminocarbonylation.** Reaction conditions: aryl halide (1.0 mmol), amines (1.5 mmol), 1.0 Fe$_2$O$_3$-O$_{vac}$ (80 mg), Et$_3$N (2.0 equiv.), CO (1.0 MPa), 1,4-dioxane (2.0 mL), 160 °C, 24 h, isolated yields; [a]36 h; [b]48 h; [c]72 h; **3a,** **3v, 3x, 3y, 3z** and **3aa** were produced using iodobenzene; bromobenzene; 1-fluoro-4-iodobenzene; 1-iodo-4-isocyanobenzene; 1-bromo-4-isocyanobenzene and 1-bromo-4-fluorobenzene as substrates respectively.

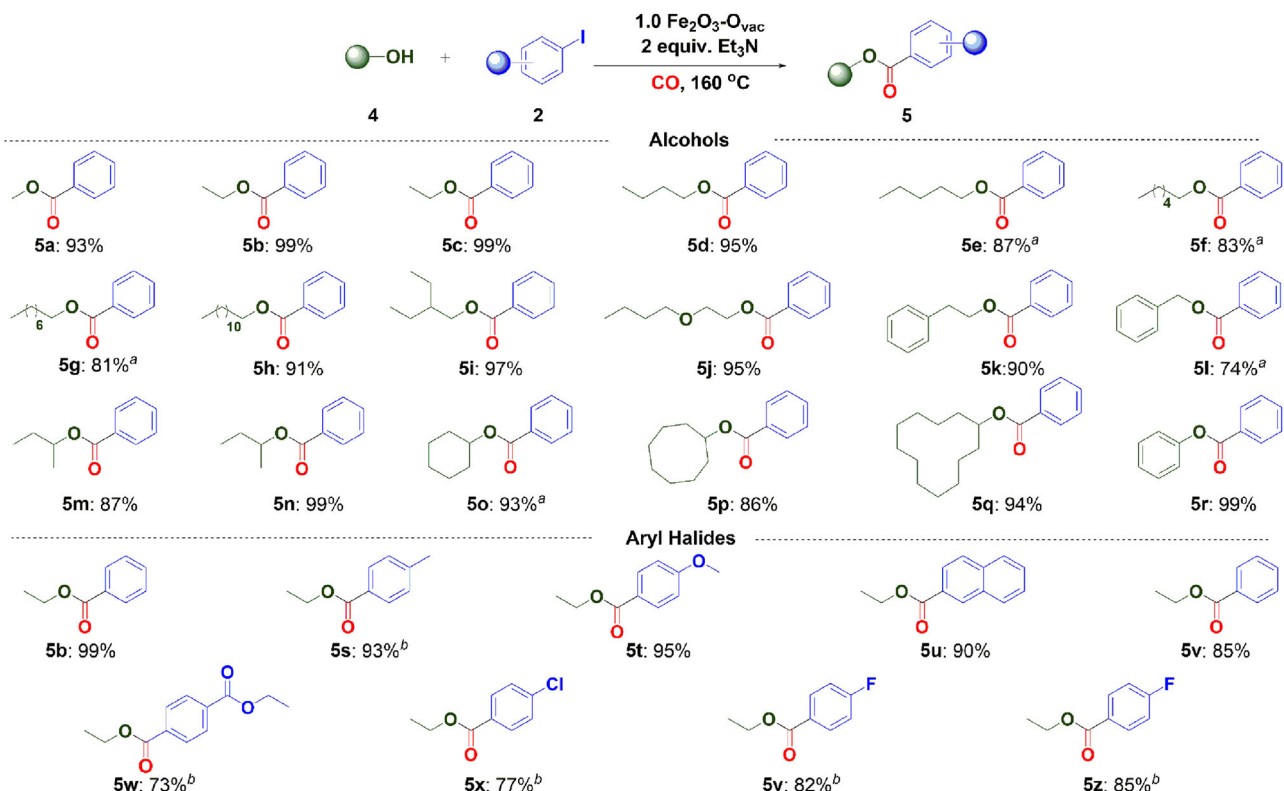

**Fig. 7 | Substrate scope of alkoxycarbonylation.** Reaction conditions: aryl halide (1.0 mmol), alcohols (3.0 mL), 1.0 Fe$_2$O$_3$-O$_{vac}$ (80 mg), Et$_3$N (2.0 equiv.), CO (1.0 MPa), 160 °C, 48 h, isolated yields; [a]Yields were determined by $^1$HNMR with triphenylmethane as internal standard; [b]72 h; **5b, 5v, 5y** and **5z** were synthesized using iodobenzene; bromobenzene; 1-fluoro-4-iodobenzene and 1-fluoro-4-bromobenzene as substrates respectively.

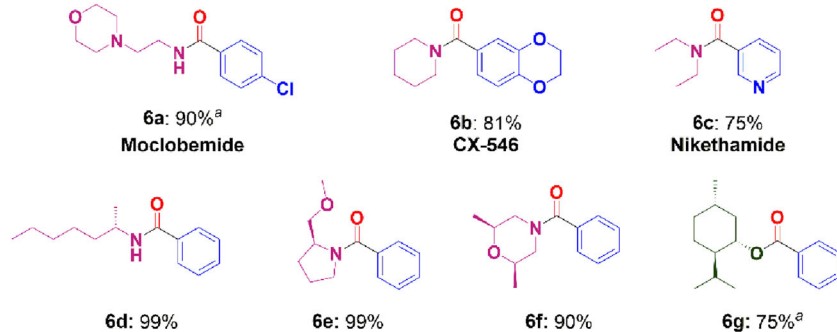

**Fig. 8 | The synthesis of pharmaceuticals and chiral compounds.** Reaction conditions: aryl halide (1.0 mmol), amines (1.5 mmol), alcohols (3.0 mL), 1.0 Fe$_2$O$_3$-O$_{vac}$ (80 mg), Et$_3$N (2.0 equiv.), CO (1.0 MPa), 160 °C, 48 h, isolated yields; [a]72 h.

into the desired product with 85% yield with the maintenance of the F substitute (**5z**).

To validate the practicality of this methodology, the synthesis of pharmaceuticals and chiral compounds was explored via carbonylation by the 1.0 Fe$_2$O$_3$-O$_{vac}$ system (Fig. 8). The antidepressant drug Moclobemide (4-chloro-*N*-(2-morpholinoethyl) benzamide)[56] (**6a**), was prepared by the aminocarbonylation of 1-chloro-4-iodobenzene with 2-morpholinoethanamine in 90% yield. Moreover, CX-546 (1-(1,4-benzodioxan-6-ylcarbonyl)piperidine) (**6b**), which was produced to treat schizophrenia[56], was also effectively synthesized with an 81% yield utilizing 6-iodo-1,4 benzodioxane and piperidine as substrates. Nikethamide (**6c**), a respiratory stimulant[56], could also be synthesized by 3-iodopyridine and diethylamine with 75% yield. Furthermore, the carbonylation reaction with iodobenzene also involved the use of a few chiral amines and alcohols. Using (*R*)-hexan-2-amine, (*R*)−2-(methoxymethyl)pyrrolidine, (2*S*,6*R*)-2,6-dimethylmorpholine and (1*R*,2*S*,5*R*)-2-isopropyl-5-methylcyclohexan-1-ol as substrates, the corresponding products were obtained in 75–99% yields (Figs. 8, 6d–g). These findings demonstrated the applicability of 1.0 Fe$_2$O$_3$-O$_{vac}$ catalyst for the valorization of biologically important and complex molecules.

## Discussion

In summary, a series of Fe$_2$O$_3$-O$_{vac}$ catalysts with varying amounts of O$_{vac}$ were synthesized, which was used to the carbonylation of aryl halides and amines/alcohols with CO. The ideal 1.0 Fe$_2$O$_3$-O$_{vac}$ system displayed excellent activity and selectivity for the synthesis of carbonylated chemicals (60 examples), including drugs and chiral related molecules, via aminocarbonylation and alkoxycarbonylation. Characterizations (XPS, EPR, TGA, Raman and XAFS) revealed the formation of O$_{vac}$ and variation of Fe sites on Fe$_2$O$_3$-O$_{vac}$ triggered by O$_{vac}$. The experimental studies and DFT calculations verified that the catalytic performance of the carbonylation were significantly improved because the selective combination of these three Fe sites can catalyze the elementary step of PhI activation, CO insertion and C-N/C-O coupling efficiently. This work provided a concept to design NMCs catalysts and study the origin of improved catalytic performance for multistep reactions.

## Methods
### General considerations

All solvents and chemicals, unless otherwise noted, were obtained commercially, and were used as received without further purification. All glassware was dried before using. Analytical thin layer chromatography was performed using pre-coated Jiangyou silica gel HSGF254 (0.2 mm ± 0.03 mm). Flash chromatography was performed using silica gel 60, 0.063–0.2 mm, 200–300 mesh (Jiangyou, Yantai) with the indicated solvent system.

### Characterizations

GC-MS analysis was in general recorded on an Agilent 5977 A MSD GC-MS.

Fourier transform infrared (FT-IR) spectrum[57] were recorded with a Bruker VERTEX 70FTIR spectrometer.

The in situ DRIFTS[58] of samples were analyzed by a Bruker VERTEX 70 FTIR spectrometer and used for the identification of IR absorbance in the mid-IR region (400–4000 cm$^{-1}$). It was equipped with liquid-nitrogen-cooled MCT detector and low-volume gas cell (8.7 mL) with a 123 mm path length and KBr windows. The sample was pretreated at 30 °C for 10 min under CO (5 mL/min) and heated to 180 °C.

Raman spectroscopy[57] (LabRAM HR Evolution Raman spectrometer) was performed by employing a 532 nm laser beam.

The liquid nuclear magnetic resonance spectra (NMR) were recorded on a Bruker AvanceTM III 400 MHz in deuterated chloroform unless otherwise noted. Data are reported in parts per million (ppm) as follows: chemical shift, multiplicity (s = singlet, d = doublet, t = triplet, q = quartet, quint = quintet, m = multiplet, dd = doublet of doublet and br = broad signal), coupling constant in Hz and integration[59].

XRD measurements[60] were conducted were conducted by a STADIP automated transmission diffractometer (STOE) equipped with an incident beam curved germanium monochromator selecting CuKα1 radiation and a 6° position sensitive detector (step size: 0.014°, step time: 25.05 s). The XRD patterns were scanned in the 2θ range of 10–90°.

XPS measurements[60] were carried out by a VG ESCALAB 210 instrument equipped with a dual Mg/Al anode X-ray source, a hemispherical capacitor analyzer, and a 5 keV Ar$^+$ ion gun. All spectra were recorded by using AlKa (1361 eV) radiation. The electron binding energy was referenced to the C1s peak at 284.8 eV.

The thermal properties[60] of Fe$_2$O$_3$ catalysts were evaluated using a METTLER TOLEDO simultaneous thermal analyzer over the temperature range from 30 to 800 °C under nitrogen atmosphere (20 mL/min) with a heating rate of 5 °C/min. In H$_2$-TGA analyses, ~5 mg of catalyst was used, and the change in weight was recorded in the temperature range of 30–850 °C at a heating rate of 5 °C min$^{-1}$ under 5% H$_2$/Ar flow of 20 mL min$^{-1}$.

High-resolution TEM analysis[60] was carried out on a JEM 2010 operating at 200 KeV. The catalyst samples after pretreatment were dispersed in ethanol, and the solution was mixed ultrasonically at room temperature. A part of solution was dropped on the grid for the measurement of TEM images.

EXAFS experiments[61] were performed at the Beijing Synchrotron Radiation Facility (BSRF) in Beijing Institute of High Energy Physics, Chinese Academy of Sciences with a storage ring energy of 2.5 GeV and a beam current between 150 and 250 mA. The Fe K-edge absorbance of powder catalysts was measured in transmission geometry at room temperature. EXAFS data analysis was carried out using ifeffit analysis programs (http://cars9.uchicago.edu/ifeffit/). Radial distribution

functions were obtained by Fourier-transformed k3-weighted X function.

EPR spectra[61] were recorded at room temperature on a Bruker cw spectrometer EMX-PLUS (X-band, $v \approx 9.8\,GHz$) with a microwave power of 20 mW, a modulation frequency of 100 kHz and modulation amplitude of up to 1 G, the usage of sample was 10 mg.

Mössbauer measurements[38] were performed using a conventional constant acceleration type spectrometer in transmission geometry in the temperature range from 6 to 300 K. Absorbers were prepared in powder form (10 mg of natural Fe cm$^{-2}$). The γ-ray source is a commercial 25 mCi $^{57}$Co in a palladium matrix. The driver velocity was calibrated using sodium nitroprusside powder and all isomer shifts were quoted relative to the α-Fe foil at room temperature.

The element type and content of the catalyst were determined by inductively coupled plasma optical emission spectrometry (ICP-OES)[58]. Preparation of test sample: 20 mg sample was dissolved with a mixture of concentrated nitric acid and hydrochloric acid, and heated until the sample was completely dissolved, and then the clarified transparent solution was quantitatively transferred to a volumetric flask.

$O_2$-TPD was performed on a chemisorption analyzer equipped with a thermal conductivity detector (TCD)[57]. The chemisorption analyzer was TP-5080D from Tianjin Xianquan Industrial and Trading Co., Ltd. The weighed sample (100 mg) was pretreated at 300 °C for 1 h under He (40 mL/min) and cooled to 30 °C. The $O_2$ gas (30 mL/min) was introduced instead of He gas at this temperature for 1 h to ensure the saturation adsorption of $O_2$. The sample was then purged with He for 1 h (40 mL/min) until the signal returned to the baseline as monitored by a TCD. The desorption curve of $O_2$ was acquired by heating the sample from 30 to 600 °C at 10 °C/min under He with the flow rate of 40 mL/min.

$H_2$-TPR was performed on a chemisorption analyzer equipped with a TCD[57]. The chemisorption analyzer was TP-5080D from Tianjin Xianquan Industrial and Trading Co., Ltd. The weighed sample (10 mg) was pretreated at 300 °C for 1 h under He (40 mL/min) and cooled to 30 °C. The $H_2/N_2$ gas ($H_2$: 5 wt%, 30 mL/min) was introduced instead of He for 1 h until the signal returned to the baseline as monitored by a TCD. The reduction curve of $H_2$ was acquired by heating the sample from 30 to 800 °C at 10 °C/min under $H_2/N_2$ gas with the flow rate of 30 mL/min.

## Synthesis of Fe$_2$O$_3$

500 mg anhydrous ferric chloride and 2 g CTAB (cetyltrimethyl ammonium bromide) were dissolved in 60 mL deionized. After the mixture became clear solution, the above mixture was transferred into a 100 mL Teflon-lined stainless-steel autoclave and heated to 120 °C for 24 h and then cooled to room temperature naturally. The resulting product was collected by filtration, washed several times with deionized water and absolute ethanol, and then dispersed in absolute ethanol and dried at 80 °C in air overnight. The sample was labeled as Fe$_2$O$_3$.

## Synthesis of Fe$_2$O$_3$-O$_{vac}$

320 mg Fe$_2$O$_3$ was added to a Shrek tube followed by exchange with Ar, and then 20 mL mixture of $H_2O$ and EtOH ($V(H_2O)/V(EtOH) = 1/4$) were added with 10 min magnetic stirring at room temperature. Then different amount of NaBH$_4$ was added into the Shrek tube and maintained for 20 min. The product was washed with absolute ethanol three times and dried in vacuum at 75 °C for 6 h to gain a series of Fe$_2$O$_3$-O$_{vac}$ with different amount of oxygen vacancy, which were denoted as 0.5 Fe$_2$O$_3$-O$_{vac}$, 1.0 Fe$_2$O$_3$-O$_{vac}$ and 2.0 Fe$_2$O$_3$-O$_{vac}$.

## DFT calculations

Spin-polarized density functional theory method were performed in this work as imvia the Vienna Ab initio Simulation Package (VASP)[62]. The projected augmented wave method (PAW)[63] was used to describe the interaction of electron and ion. A Hubbarf $U$ ($U_{eff} = 4\,eV$) was used to treat the strong correlated electrons of the localized Fe 3d-orbital[64,65]. The electron exchange and correlation energies were calculated within the generalized gradient approximation method (GGA) using the Perdew-Burke-Ernzerhof (PBE) functional[66,67]. To make sure that the energy difference is less than $10^{-4}\,eV$ and the force per atom is less than 0.03 eV/Å, Gaussian smearing (0.02 eV) was used. And Gamma k point was adopted for sampling at Brillouin zone. The kinetic energy cutoff was set up to 500 eV and dispersion correction was considered by using DFT-D3 method with Becke-Jonson damping [68]. The vacuum layer between the periodically repeated slabs was set as 20 Å.

The adsorption energy ($E_{ads}$) of adsorbate (X) is obtained from the equation $E_{ads} = E_{X/slab} - E_{slab} - E_X$, where $E_{X/slab}$ is the total energy after adsorption, $E_{slab}$ is the total energy of the clean surface, $E_X$ is the total energy of the free adsorbate (X) in a $20 \times 15 \times 15$ cubic box; and therefore, the more negative $E_{ads}$, the stronger of the interactions between the adsorbates and surface; and the opposite number of $E_{ads}$ is regarded as the desorption energy $E_{des}$. For reactions, the climbing-image nudged elastic band (CI-NEB) method[69] was adopted to search the transition states (TS) and the vibrational frequency analysis was also processed to verify the authentic transition state with only one imaginary frequency. The reaction barrier ($E_a$) is defined as $E_a = E_{TS} - E_{IS}$ and the reaction energy ($E_r$) is defined as $E_r = E_{FS} - E_{IS}$, where $E_{IS}$, $E_{FS}$ and $E_{TS}$ are the total energies of the initial, final and transition states, respectively. Zero point energy (ZPE) correction was included in all energies.

Two models were adopted to clarify the activities of flawed and normal Fe$_2$O$_3$(104): Fe$_2$O$_3$-O$_{vac}$, Fe$_2$O$_3$ (Supplementary Fig. 2a, b). During simulation, the bottom 32 Fe and 48 O atoms were fixed.

## General procedure for the aminocarbonylation of aryl halides

Typical procedure for carbonylation of aryl iodides with amines and CO. A mixture of aryl iodides (1.0 mmol), amines (1.5 mmol), catalysts (80 mg), Et$_3$N (2.0 mmol) and dioxane (2 mL) were added a glass tube which was placed in an 80 mL autoclave. Then the autoclave was purged and charged with CO (1.0 MPa). The reaction mixture was stirred at 160 °C for 24 h. After the reaction finished, the autoclave was cooled to room temperature and the pressure was carefully released. Subsequently, the reaction mixture was diluted with 5 mL of methanol for analysis by GC-MS. The crude reaction mixture was concentrated by rot-vap and purified by column chromatography on a silica gel column to give the desired products.

## General procedure for the alkoxycarbonylation of aryl halides

Typical procedure for carbonylation of aryl iodides with alcohols and CO. A mixture of aryl iodides (1.0 mmol), alcohols (3 mL), catalysts (80 mg) and Et$_3$N (2.0 mmol) were added a glass tube which was placed in an 80 mL autoclave. Then the autoclave was purged and charged with CO (1.0 MPa). The reaction mixture was stirred at 160 °C for 48 h. After the reaction finished, the autoclave was cooled to room temperature and the pressure was carefully released. Subsequently, the crude reaction mixture was concentrated by rot-vap and purified by column chromatography on a silica gel column to give the desired products.

## Data availability

Data supporting key conclusions of this work are contained within the paper and Supplementary Information. All raw data used in the current study are available from the corresponding author under request. Source data are provided with this paper.

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

## Acknowledgements

We thank the National Natural Science Foundation of China (No. 21925207, 22102195, 22102199), the Natural Science Foundation of Gansu Province, China and the Major Project of Gansu Province, China (No. 21JR7RA096, 21ZD4WA021), and Key Program of the Lanzhou Institute of Chemical Physics, CAS (No. KJZLZD-1, KJZLZD-2) for the financial supports. We thank Prof. Dr. Lirong Zheng (Institute of High Energy Physics, CAS) for helping us to perform the EXAFS (1W1B BSRF) analysis.

## Author contributions

X.C. and F.S. conceived the idea, supervised the project. S.L. designed and performed experiments. T.L. designed and carried out the DFT calculations. X.D. performed XANES and EXAFS analyses. H.M. and B.W. supported the XRD, O2-TPD and H2-TPR analysis. X.C., S.L., and T.L. prepared the manuscript. All authors discussed the results and assisted during manuscript preparation.

## Competing interests

The authors declare no competing interests.
