## [Peer Review File · Nature Communications]

Constructing multiple active sites in iron oxide catalysts for improving carbonylation reactionsREVIEWER COMMENTS

Reviewer #1 (Remarks to the Author):

The manuscript is well written and organized. However, its novelty is quite limited. Indeed, oxygen defect engineering is well-known and extensively reported strategy to control/improve catalytic efficiency of metal oxides in various chemical reactions. The insertion of oxygen vacancies into iron oxides and related compounds represents the frequent approach to control the yield and selectivity of such catalysts (e.g. Wang et al. ACS Catal. 2021; Guo et al. J. Mater. Chem. A 2019).

Due to the limited novelty I do not recommend the manuscript for publication in Nature Communications. Other issues, which should be addressed before further consideration in any journal:

- The manuscript does not deal with the polymorphous character of iron(III) oxide (hematite vs maghemite vs epsilon-Fe₂O₃ vs beta-Fe₂O₃). The particular studied polymorph should be specified and possible presence/formation of other polymorphs addressed.
- The overall stoichiometry of iron(III) oxides prepared via the reduction approach should be precisely calculated through the formulae of Fe₂O_{3-x}. This is simply doable exploiting XPS, ICP and other techniques.
- High resolution XPS data should be discussed in greater detail. HR O1s XPS spectra of iron(III) oxides differing in the vacancy concentrations should be also analysed in terms of the content of surface hydroxyl groups. Their possible effect on the catalytic efficiency should be properly discussed and their role in the reaction mechanism explored.
- The concentration of other metals in the starting material should be quantified and their possible effect addressed.
- The change of the binding energies for the lattice oxygen (the peak narrowing in the particular O 1s XPS spectra) should be addressed.
- TGA curves of the vacancy-containing Fe₂O₃ sample is completely the same with the pristine Fe₂O₃. This is little bit surprising with respect to the differences in relevant XPS spectra. The high temperature TG experiments in hydrogen atmosphere would be more relevant to evaluate the amount of vacancies and the level of samples non-stoichiometry.

Reviewer #2 (Remarks to the Author):

Developing efficient, inexpensive and robust heterogeneous catalysts for coupling reactions based on non-noble metal is greatly important. The manuscript by Cui and co-workers entitled as "Constructing combinatorial site iron catalysts triggered by oxygen vacancy for highly efficient carbonylation reaction" reported a surface-engineered oxygen vacancies (Ovac) induced Fe₂O₃ catalyst with three different Fe sites around one oxygen vacancy and exhibited significant catalytic performance for the carbonylation of various aryl halides and amines/alcohols with CO.

The materials have been well characterised using all the relevant analytic techniques such as XPS, EPR, TGA, Raman and XAFS. Also, the authors provided various control experiments and density functional theory (DFT) calculations to demonstrate the selective combination of these three Fe sites and their catalytic performances. The substrate scope is quite broad. The work is of interest to the materials chemistry community as well as in synthetic transformations for sustainable developments. This reviewer recommends the work for publication in Nature Communications only after some major changes.

1. The authors need to provide the XRD matching with the JCPDS data (JCPDS no. 890599)? What is the percentage of matching? Normally, any reaction condition can generate oxygen vacancies. Also, any other kind of defects also plays a role in enhancing the intensity of Ovac in XPS, EPR or Raman spectra. Please check out that. In general, the reaction of iron chloride and sodium borohydride form iron nanoparticles. The contribution of iron needs to be included in the results.
2. In TEM image, fringe spacing is not clear. It looks amorphous. The image and fringe spacing should be zoomed and clear.
3. Did the concentration of samples during EPR measurement in each sample kept same? As the vacancies generally do the broadening of the peak. Hence, the higher intensity might have arisen from other sources. Check out this.
4. In Raman spectra, one extra peak at 400 cm^{-1} appeared in Fe_2O_3 and 0.5 Fe_2O_3 -Ovac. Authors should describe the peak position.
5. What is the role of triethylamine?
6. Generally, the activation or oxidative addition of haloarenes is reported with iron complexes in lower oxidation states (0 or -1). In the present system, the oxidation states of the iron are either +2 or +3. Therefore, the expected mechanistic path will be different from a transition metal process. Alternatively, a radical mechanism is very probable, and the amine can act as a source of a sacrificial reductant. The authors need to elucidate the probable mechanism of the reaction with further controlled experiments.
7. The authors need to provide the ICP-OES data to exclude the role of other metals such as copper, nickel, palladium etc.

Reply to referee 1:

The manuscript is well written and organized. However, its novelty is quite limited. Indeed, oxygen defect engineering is well-known and extensively reported strategy to control/improve catalytic efficiency of metal oxides in various chemical reactions. The insertion of oxygen vacancies into iron oxides and related compounds represents the frequent approach to control the yield and selectivity of such catalysts (e.g. Wang et al. ACS Catal. 2021; Guo et al. J. Mater. Chem. A 2019).

Reply: We agree with your comments about the importance of the oxygen defect engineering to regulate the surface charge redistribution and to improve the catalytic efficiency in chemical reactions.

We carefully checked the literatures mentioned here. Wang et al. (Ref: **ACS Catal.** 2021, 11, 11256) reported the oxygen vacancy-rich ultrathin 2D Fe₃O₄ on peroxymonosulfate activation. The active sites were Fe³⁺ and Fe²⁺ of Fe₃O₄ rather than that generated by oxygen vacancies. Moreover, the oxygen vacancies promoted the electron transfer and the cycle of Fe³⁺/Fe²⁺, enhancing the catalytic activity. Guo et al. (Ref: **J. Mater. Chem. A**, 2019, 7, 21704) reported the oxygen vacancies on the surface of CoFe₂O₄ nanosheet could reduce adsorption energy of H₂O and exhibited desired overall water splitting activity, but only one Fe site was involved in the reaction. Although the catalytic performance was enhanced by the formation of oxygen vacancies as mentioned above, more attentions are paid on the electronic influence between O_{vac} and adjacent single metal site and normally only one O_{vac}-generated Fe site was studied in the reaction. However, the formation of one oxygen vacancy normally causes the micro-environmental change of multiple metal sites. Their catalytic differences and synergistic effect during the reaction were rarely studied and elusive.

As we all know, the subtle variation in the active site architecture can affect catalytic performance significantly and even change the reaction pathway, thereby different elementary steps for a multistep reaction might be selectively determined by one of multiple metal sites. Herein, three different Fe sites (denoted as **Fe1**, **Fe2** and **Fe3** in the manuscript) are formed around one oxygen vacancy, and the selective combination of these three Fe sites catalyzes different elementary steps of aminocarbonylation reaction: PhI activation (**Fe1** and **Fe3**), CO insertion (**Fe1** and **Fe2**), C-N coupling (**Fe3**, **Fe1**

and **Fe₂**), respectively. This work revealed the catalytic differences of the multiple metal sites around O_{vac} and improved catalytic performance was achieved by combinatorial catalysis of these multiple metal sites.

(1) The manuscript does not deal with the polymorphous character of iron(III) oxide (hematite vs maghemite vs epsilon-Fe₂O₃ vs beta-Fe₂O₃). The particular studied polymorph should be specified and possible presence/formation of other polymorphs addressed.

Reply: Thank you very much for your suggestion. As shown in Supplementary Fig. 3, the diffraction peaks of Fe₂O₃, 0.5 Fe₂O₃-O_{vac} and 1.0 Fe₂O₃-O_{vac} could be well indexed to the standard XRD pattern of hematite Fe₂O₃ (PDF #: 890599). The corresponding discussions were given in revised manuscript of page 3, lines 89-91: "The XRD patterns (Supplementary Fig. 3) exhibited that Fe₂O₃ samples were formed with a high crystalline structure where (104) plane was dominant, which could be well indexed to the standard XRD pattern of hematite Fe₂O₃ (PDF #: 890599)^[38]."

(2) The overall stoichiometry of iron(III) oxides prepared via the reduction approach should be precisely calculated through the formulae of Fe₂O_{3-x}. This is simply doable exploiting XPS, ICP and other techniques.

Reply: Thank you very much for your suggestion. The overall stoichiometry of iron(III) oxides have been precisely calculated through the formulae of Fe₂O_{3-x} by the ICP and XANES, the related stoichiometry of 0.5 Fe₂O₃-O_{vac} and 1.0 Fe₂O₃-O_{vac} were Fe₂O_{2.99} and Fe₂O_{2.96}. The corresponding discussions were given in revised manuscript of page 5, lines 163-164: "The related stoichiometry of 0.5 Fe₂O₃-O_{vac} and 1.0 Fe₂O₃-O_{vac} were Fe₂O_{2.99} and Fe₂O_{2.96}."

(3) High resolution XPS data should be discussed in greater detail. HR O1s XPS spectra of iron(III) oxides differing in the vacancy concentrations should be also analysed in terms of the content of surface hydroxyl groups. Their possible effect on the catalytic efficiency should be properly discussed and their role in the reaction mechanism explored.

Reply: Thank you very much for your suggestion. High resolution XPS data has been further discussed. And the corresponding discussions were given in revised manuscript of page 4, lines 103-114: "X-ray photoelectron spectroscopy (XPS) was conducted to study the composition of surface oxygen species and the charge distribution. All XPS spectra were charge corrected and

referenced with adventitious carbon (284.8 eV). The survey spectra of three Fe₂O₃ samples were shown in Supplementary Fig. 6, which clearly revealed the coexistence of Fe, O, and C elements of three samples. The density of oxygen vacancy on these catalysts can also be deduced from O 1s spectra. As shown in Fig. 3a, three peaks can be deconvoluted from the O 1s profiles. The peaks at 529.8, 531.4, and 533.0 eV can be ascribed to surface lattice oxygen (O_L), surface O_{vac} and other weakly bound oxygen species such as adsorbed molecular water and hydroxyl groups (O_{OH})^[40]. The density of O_{vac} could be assumed as $O_{vac}/(O_{vac} + O_L + O_{OH})$. As shown in Fig. 3a and Supplementary Table 3, the density of O_{vac} was in the following order: 1.0 Fe₂O₃-O_{vac} (26%) > 0.5 Fe₂O₃-O_{vac} (18%) > Fe₂O₃ (13%), suggesting that the content of oxygen vacancies was gradually increased using larger NaBH₄ amounts.”.

The content of surface hydroxyl groups was also analysed in Supplementary Table 3, the results showed that the content of surface hydroxyl groups of Fe₂O₃, 0.5 Fe₂O₃-O_{vac} and 1.0 Fe₂O₃-O_{vac} were almost similar, which revealed the catalytic efficiency was not affected by the surface hydroxyl groups. The corresponding discussions were given in revised manuscript of page 4, lines 116-119: “The content of surface hydroxyl groups was also analysed in Supplementary Table 4, which showed that the content of surface hydroxyl groups of Fe₂O₃, 0.5 Fe₂O₃-O_{vac} and 1.0 Fe₂O₃-O_{vac} were almost similar, which revealed the catalytic efficiency was not affected by the surface hydroxyl groups.”.

Moreover, the role of surface hydroxyl groups in the reaction mechanism have been explored by DFT calculations. According to the Supplementary Fig. 16, the activation barriers for PhI dissociation on the surface without hydroxyl groups (0.54 eV) were almost the same with that on the surface with hydroxyl groups (0.55 eV). However, the reaction energy (-0.55 eV) on the surface without surface hydroxyl groups was lower than that (-0.06 eV) on the surface with hydroxyl groups, indicating that PhI dissociation on the surface decorated with hydroxyl groups was more exergonic. Therefore, the existence of surface hydroxyl groups was not beneficial for PhI decomposition.

We have added the corresponding discussions in revised manuscript of page 10, lines 270-274: “The hydroxyl groups, which might exist on the surface, had little effect on the energy barrier (0.55 eV on the surface with hydroxyl groups) but increased the reaction energy (-0.06 eV) of PhI dissociation,

indicating that surface hydroxyl groups were not beneficial for C-I bond decomposition (**IM1'**→**TS1'**→**IM2'**, Supplementary Fig. 16). Therefore, PhI activation preferred to occurred on the vacant site without surface hydroxyl groups."

Supplementary Fig. 16 Structures of iodobenzene was adsorbed at $Fe_2O_3(104)-O_{vac}$ surface with O_{vac} (**IM1**), $Fe_2O_3(104)-O_{vac}$ with hydroxyl (**IM1'**), and normal $Fe_2O_3(104)$ (**IM1''**). Structural evolution of PhI decomposition on $Fe_2O_3(104)-O_{vac}$ without hydroxyl (**Reaction 1**), $Fe_2O_3(104)-O_{vac}$ with hydroxyl (**Reaction 1'**), normal $Fe_2O_3(104)$ (**Reaction 1''**).

(4) The concentration of other metals in the starting material should be quantified and their possible effect addressed.

Reply: Thank you very much for your suggestion. The concentrations of other metals in the starting material and $Fe_2O_3-O_{vac}$ catalyst have been quantified by the ICP-OES in Supplementary Table 1. The results from ICP-OES analysis for the starting material ($FeCl_3$) and $Fe_2O_3-O_{vac}$ catalyst revealed that the content of other metals such as Cu, Ni and Pd was below the limit of detection. Moreover, $Fe_2O_3-O_{vac}$ catalysts containing 250 ppm of Cu, Ni and Pd

were prepared and their catalytic activity were studied respectively. As shown in Supplementary Table 2, the addition of Cu and Ni metals in the Fe₂O₃-O_{vac} catalyst decreased the catalytic activity, while the Pd metal exhibited negligible effect. The corresponding discussions have been added in revised manuscript of page 3, lines 83-85: "As shown in Supplementary Table 1, ICP-OES analysis for the starting material (FeCl₃) and Fe₂O₃-O_{vac} catalyst revealed that the content of other metals such as Cu, Ni and Pd was below the limit of detection." and page 7, lines 198-202: "To exclude the effect of these metals in the catalytic performance, Fe₂O₃-O_{vac} catalysts containing 250 ppm of Cu, Ni and Pd were prepared and their catalytic activity were studied respectively. As shown in Supplementary Table 2, the addition of Cu and Ni metals in the Fe₂O₃-O_{vac} catalyst decreased the catalytic activity, while the Pd metal exhibited negligible effect (entries 5-7)."

(5) The change of the binding energies for the lattice oxygen (the peak narrowing in the particular O 1s XPS spectra) should be addressed.

Reply: Thank you very much for your suggestion. We have addressed the mentioned problems in Fig 3(a) in the revised manuscript.

(6) TGA curves of the vacancy-containing Fe₂O₃ sample is completely the same with the pristine Fe₂O₃. This is little bit surprising with respect to the differences in relevant XPS spectra. The high temperature TG experiments in hydrogen atmosphere would be more relevant to evaluate the amount of vacancies and the level of samples non-stoichiometry.

Reply: Thank you very much for your comments. The high temperature H₂-TG experiments had been performed and the results were shown in the Supplementary Fig. 9. Two separated reduction stages could be distinguished, the first region between 250 and 500 °C corresponds to the reduction of Fe(III) to Fe(II), and the oxygen vacancies are mainly produced in this region. The weight loss decreases from 1.0 Fe₂O₃-O_{vac} to Fe₂O₃, indicating that the more oxygen vacancies formed in 1.0 Fe₂O₃-O_{vac} sample. The corresponding discussions were given in page 4, lines 136-141: "The weight losses observed in the H₂-TGA analysis resulted from the desorption of water formed by the reduction reaction (Fe₂O₃ + H₂ → Fe²⁺/Fe⁰ + H₂O) [44] in the Supplementary Fig. 9, and two separated reduction stages can be distinguished. The first region between 250 and 500 °C corresponded to the reduction of Fe(III) to Fe(II) where the oxygen vacancies were mainly produced. The weight loss decreased in

sequence of 1.0 Fe₂O₃-O_{vac}, 0.5 Fe₂O₃-O_{vac} and Fe₂O₃, indicating the more oxygen vacancies were formed in 1.0 Fe₂O₃-O_{vac} sample.”.

Reply to referee 2:

(1) The authors need to provide the XRD matching with the JCPDS data (JCPDS no. 890599)? What is the percentage of matching? Normally, any reaction condition can generate oxygen vacancies. Also, any other kind of defects also plays a role in enhancing the intensity of O_{vac} in XPS, EPR or Raman spectra. Please check out that. In general, the reaction of iron chloride and sodium borohydride form iron nanoparticles. The contribution of iron needs to be included in the results.

Reply: Thank you very much for your suggestions. As shown in Supplementary Fig. 3, the diffraction peaks of Fe₂O₃, 0.5 Fe₂O₃-O_{vac} and 1.0 Fe₂O₃-O_{vac} could be well indexed to the standard XRD pattern of hematite Fe₂O₃ (PDF #: 890599) with 100% matching percentage. The corresponding discussions were given in revised manuscript of page 3, lines 89-91: “The XRD patterns (Supplementary Fig. 3) exhibited that Fe₂O₃ samples were formed with a high crystalline structure where (104) plane was dominant, which could be well indexed to the standard XRD pattern of hematite Fe₂O₃ (PDF #: 890599) [38].”.

Similar methods of preparation metal oxides with oxygen vacancies by NaBH₄ reduction treatment have been achieved from references (*Adv. Energy Mater.* **2014**, 4, 1400696, *J. Mater. Chem. A*, **2014**, 2, 6727, *Nanoscale*, **2019**, 11,12477, *Small* **2022**, 18, 2107938), and oxygen vacancies have been revealed by XPS, EPR and Raman. The related references have been added in the revised manuscript, ref. 34, 35, 36, 37.

Combined the characterizations of ICP-OES, XRD and XPS survey, it can be found that the catalysts were hematite Fe₂O₃, and only contained Fe and O elements of the three samples. Based on the work by Chen, S. et al. (ref:42), the g value, which equals to 2.003, was ascribed to oxygen vacancy.

Some control experiments have been conducted to reveal the contribution of iron nanoparticles on the catalytic performance. As shown in the Supplementary Table 2, much lower yields of the desired product were achieved when the iron nanoparticles and physical mixture of iron nanoparticles and 1.0 Fe₂O₃-O_{vac} catalysts were used as catalyst (entries 5-7).The

corresponding discussions were given in page 6 and page 7, lines 196-198: “Moreover, control experiments where the iron nanoparticles and physical mixture of iron nanoparticles and 1.0 Fe₂O₃-O_{vac} catalysts were conducted, much lower yields of amide were obtained after adding iron nanoparticles (entries 2-4, Supplementary Table 2).”.

The ⁵⁷Fe Mössbauer spectroscopy was used to explore the oxidation state of Fe atoms. The Mössbauer spectroscopy of Fe₂O₃, 0.5 Fe₂O₃-O_{vac} and 1.0 Fe₂O₃-O_{vac} revealed only single sextet indicating magnetically ordered state (Supplementary Fig. 4). The high values of hyperfine field B_{hf} (Supplementary Table 3) obtained for the all samples were associated with the hematite Fe₂O₃. Combined the results of above control experiments and Mössbauer spectroscopy, no iron nanoparticles were formed during catalyst preparation. The corresponding discussions were given in revised manuscript of page 3, lines 91-95: “⁵⁷Fe Mössbauer spectroscopy is a very useful technique to explore the local magnetic behaviour as well as the oxidation state of Fe atoms, the transmission mössbauer spectra at room temperature for Fe₂O₃, 0.5 Fe₂O₃-O_{vac} and 1.0 Fe₂O₃-O_{vac} (Supplementary Fig. 4) only exhibited the single of the hematite Fe₂O₃, excluding the formation of the iron nanoparticles (Supplementary Table 3) [39], which was consistent with the XRD analysis.”.

(2) In TEM image, fringe spacing is not clear. It looks amorphous. The image and fringe spacing should be zoomed and clear.

Reply: Thank you very much for your suggestions. The image and fringe spacing in TEM image have been zoomed in and cleared in the revised Fig. 2 and Supplementary Fig. 13.

(3) Did the concentration of samples during EPR measurement in each sample kept same? As the vacancies generally do the broadening of the peak. Hence, the higher intensity might have arisen from other sources. Check out this.

Reply: Thank you very much for your suggestions. The concentration of samples during EPR measurement in each sample kept the same and the usage of sample was 10 mg. Other sources affecting peak intensity include the EPR measurement conditions and catalyst composition. EPR spectra were recorded under the same conditions with a microwave power of 20 mW, a modulation frequency of 100 kHz and modulation amplitude of up to 1 G at room temperature. The characterization of XRD and Mössbauer spectroscopy

showed that the purity of hematite phase of our catalysts. Combined EPR measurement conditions and the existing results of catalyst characterization, it can be found that the signal of EPR spectra derived from oxygen vacancies. Moreover, the normalized EPR signal intensities were correlated with oxygen vacancies concentration, and significantly improved EPR intensity indicated the formation of abundant oxygen vacancies. Similar comments were achieved from reference (*Nat. Commun.* **2019**, 10, 1), which we have cited as ref. 42.

The corresponding discussions were given in “Methods” section in revised manuscript: “EPR spectra were recorded at room temperature on a Bruker cw spectrometer EMX-PLUS (X-band, $\nu \approx 9.8$ GHz) with a microwave power of 20 mW, a modulation frequency of 100 kHz and modulation amplitude of up to 1 G, the usage of sample was 10 mg.”

(4) In Raman spectra, one extra peak at 400 cm⁻¹ appeared in Fe₂O₃ and 0.5 Fe₂O₃-O_{vac}. Authors should describe the peak position.

Reply: Thank you for your kind reminder, the peak at 408.8 cm⁻¹ at Fe₂O₃ and 0.5 Fe₂O₃-O_{vac} was attributed to the Fe-O symmetric bending vibrations (E_g mode). The corresponding discussions were given in the revised manuscript of page 4 and page 5, lines 143-145: “The peaks located at 221.7 cm⁻¹ was assigned to the Fe-O symmetric stretching vibrations (A_{1g} mode), and the two peaks at about 287.9, and 408.8 cm⁻¹ were attributed to the Fe-O symmetric bending vibrations (E_g mode).”.

(5) What is the role of triethylamine?

Reply: The triethylamine was generally added to neutralize the hydrogen halide, which formed in the replacement between HNR and I. The corresponding discussions were given in the revised manuscript of page 6, lines 188-189: “The triethylamine was generally added to neutralize the hydrogen halide formed during the reaction [55].”.

(6) Generally, the activation or oxidative addition of haloarenes is reported with iron complexes in lower oxidation states (0 or -1). In the present system, the oxidation states of the iron are either +2 or +3. Therefore, the expected mechanistic path will be different from a transition metal process. Alternatively, a radical mechanism is very probable, and the amine can act as a source of a sacrificial reductant. The authors need to elucidate the probable mechanism of the reaction with further controlled experiments.

Reply: We appreciate your nice suggestions. After carefully checking the literatures, the activation of aryl halides was reported as the rate-determining step using homogeneous transition metal complexes. However, the CO migratory insertion was the rate-determining step in our 1.0 Fe₂O₃-O_{vac} system which was different from transition metal process. The corresponding discussions have been added in the revised manuscript of page 10, line 302-305: “Homogeneous complexes where transition metals were generally in lower oxidation states initiated the reaction by the rate-determined activation of aryl halides^[59,60]. However, the CO migratory insertion (**TS2**) was the rate-determining step in our 1.0 Fe₂O₃-O_{vac} system which was different from transition metal process.”

To exclude the possibility of radical mechanism, control experiments have been performed in the presence of radical scavenger, and 99% amide products were obtained when excess TEMPO was added in aminocarbonylation of iodobenzene. The corresponding results have been added in revised Supplementary Table 8. The corresponding discussions have been added in the revised manuscript of page 8, line 245-250: “Control experiments were conducted to study the possibility of radical mechanism (Supplementary Table 8). Under the standard conditions, 99% amide (**3a**) were obtained by adding 2,2,6,6-tetramethylpiperidine-1-oxyl (TEMPO, a radical scavenger) and equivalent molar amount of morpholine, excluding the radical mechanism, and we proposed the reaction process was divided into three elementary reaction steps: the PhI activation (step I), the CO insertion (step II) and the C-N coupling (step III) (Fig. 5a)^[50].”

(7) The authors need to provide the ICP-OES data to exclude the role of other metals such as copper, nickel, palladium etc.

Reply: We appreciate your nice suggestions. The ICP-OES data of copper, nickel, and palladium in the starting material (FeCl₃) and Fe₂O₃ catalysts have been added into the Supplementary Table 1. The results revealed that the concentrations of Cu, Ni and Pd was below the limit of detection. As shown in Supplementary Table 2, the addition of Cu and Ni metals in the Fe₂O₃-O_{vac} catalyst decreased the catalytic activity, while the Pd metal exhibited negligible effect. The corresponding discussions have been added in revised manuscript of page 3, lines 83-85: “As shown in Supplementary Table 1, ICP-OES analysis for the starting material (FeCl₃) and Fe₂O₃-O_{vac} catalyst revealed that the

content of other metals such as Cu, Ni and Pd was below the detection limit.” and page 7, lines 198-202: “To further exclude the effect of these metals in the catalytic performance, Fe₂O₃-O_{vac} catalysts containing 250 ppm of Cu, Ni and Pd were prepared and their catalytic activity were studied respectively. As shown in Supplementary Table 2, the addition of Cu and Ni metals in the Fe₂O₃-O_{vac} catalyst decreased the catalytic activity, while the Pd metal exhibited negligible effect (entries 5-7).”.

REVIEWERS' COMMENTS

Reviewer #2 (Remarks to the Author):

The authors have been able to address all the concerns raised by this reviewer. The quality of the manuscript has been increased with all the additional experiments suggested by this reviewer and by the other reviewer. This reviewer recommends the acceptance of the manuscript in its present form.